# Retrieval-Augmented Foundation Model Enhances Risk Prediction Using Electronic Health Records

**Saeed Shurrab** [1 2] **Mariam Al-Omari** [1] **Dana El Samad** [1] **Farah E. Shamout** [1 2]

## Abstract

Electronic Health Records (EHR) contain rich longitudinal patient information widely used for predictive modeling. However, effectively leveraging historical data remains challenging due to long trajectories, event heterogeneity, temporal irregularity, and varying relevance of past visits. Existing approaches rely on fixed windows or uniform aggregation, which may obscure clinically important signals. We introduce `EHR-RAGp`, a retrieval-augmented foundation model that dynamically integrates relevant patient history. We construct an EHR vector database via clinically relevant chunking strategies, and employ a prototype-guided retrieval module to identify and weight the most relevant historical segments for a given prediction task. Across multiple tasks, `EHR-RAGp` consistently outperforms state-of-the-art EHR foundation models.

## 1. Introduction

Healthcare systems rely on Electronic Health Record (EHR) databases to store diverse clinical events collected during patient encounters, including diagnoses, vital signs, lab results, and medications (Nasarudin et al., 2024). Recent advances in machine learning have enabled foundation models for EHR data to represent patient trajectories as chronologically ordered sequences of events, similar to sentences in natural language (Wornow et al., 2023b). Unlike natural language, clinical events are inherently irregular, sparse, and span multiple episodes of care (Zhong et al., 2025).

Existing EHR foundation models are built on assumptions that: (1) only information within a predefined time window during the encounter is relevant for outcome prediction

(Elsharief et al., 2025); and (2) a selected subset of features, such as lab results, vital signs, or diagnoses, is sufficient for prediction (Fallahpour et al., 2024; Odgaard et al., 2024). These assumptions arise from computational constraints and challenges in modeling long-range, irregular trajectories, and may lead to discarding clinically important history. In practice, a patient's state is shaped by accumulated longitudinal information, including chronic patterns and cross-encounter interactions (Braitman & Davidoff, 1996). This exposes a key limitation in current EHR models: the inability to integrate and reason over a patient's full history.

Addressing this gap requires rethinking EHR representation learning. Instead of restricting context to a fixed time window, models should retrieve and integrate information across the full patient history. We assume that retrieval offers a scalable way to incorporate long-range history without complete encoding in the model's internal state. Inspired by the success of Retrieval-Augmented Generation (RAG) in NLP (Lewis et al., 2020; Zheng et al., 2025), we treat historical EHR events as retrievable knowledge, enabling models to dynamically focus on clinically relevant context. This reframes longitudinal modeling as a retrieval-augmented problem. Accordingly, we propose *EHR-RAGp*, a retrieval-based foundation model designed to learn from and reason over a patient's full longitudinal EHR history.

**Summary of Contributions.** We propose a framework for constructing a vector database of longitudinal EHR trajectories, supported with clinically relevant chunking strategies. Additionally, we introduce a prototype-guided retrieval mechanism that integrates clinically relevant historical chunks, enabling modeling of long-range dependencies. We also introduce `EHR-RAGp`, an EHR-native foundation model, which consistently outperforms state-of-the-art EHR models across multiple clinical prediction tasks. We make our code publicly available at: https://anonymous.4open.science/r/EHR-RAGp-8777/.

## 2. Methodology

**Preliminaries.** Let $\mathcal{D} = \{\mathcal{H}_p\}_{p=1}^{\mathcal{P}}$ represent a large-scale EHR dataset of longitudinal records for $\mathcal{P}$ patients. For a given patient $p$, the history is an ordered sequence of visits:

$$\mathcal{H}_p = \{\mathcal{V}_p^{(1)}, \mathcal{V}_p^{(2)}, \ldots, \mathcal{V}_p^{(N_p)}\},$$

[1]Department of Computer Engineering, New York University Abu Dhabi, Abu Dhabi, UAE [2]Department of Biomedical Engineering, New York University, New York, United States. Correspondence to: Saeed Shurrab <saeed.shurrab@nyu.edu>.

*Proceedings of the 2nd ICML Workshop on Foundation Models for Structured Data*, Seoul, South Korea. 2026. Copyright 2026 by the author(s).

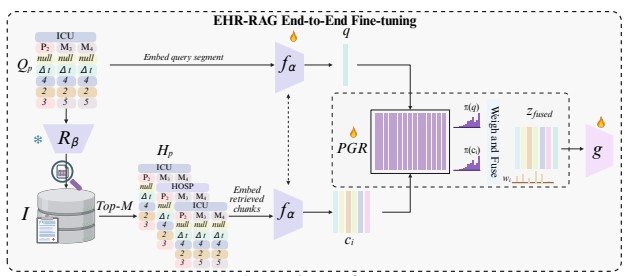

*Figure 1.* Overview of `EHR-RAGp`.

where $N_p$ denotes the number of visits. Each visit $\mathcal{V}_p^{(n)}$ consists of clinical events:

$$\mathcal{V}_p^{(n)} = \{(c_{nk}, v_{nk}, t_{nk})\}_{k=1}^{K_n}$$

where $c_{nk}$ is a clinical event (e.g., lab, vital, or medication), $v_{nk}$ is an optional value (e.g., lab result or dosage), and $t_{nk}$ is the timestamp. Events vary in type, frequency, and granularity, reflecting the irregular and heterogeneous nature of EHR data.

For an index visit $\mathcal{V}_p^{(\tau)}$, the goal is to predict an outcome with label $y_p^{(\tau)}$. The model uses this visit to retrieve relevant history. We define a **query segment** $\mathcal{Q}_p^{(\tau)} \subseteq \mathcal{V}_p^{(\tau)}$ as the portion used as input. Since full visits may exceed context limits, the query is restricted to a fixed-size segment. All prior events form the **historical context**:

$$\mathcal{H}_p^{(<\tau)} = \{\mathcal{V}_p^{(1)}, \ldots, \mathcal{V}_p^{(\tau-1)}, \ \mathcal{V}_p^{(\tau<q)}\},$$

where $\mathcal{V}_p^{(\tau<q)}$ is the prefix of the last visit before the query (see Figure S2). The model learns:

$$\gamma : \left(\mathcal{Q}_p^{(\tau)}, \mathcal{H}_p^{(<\tau)}\right) \rightarrow \hat{y}_p^{(\tau)}.$$

Instead of encoding the full history, which is computationally costly, the framework retrieves and integrates the most relevant information from $\mathcal{H}_p^{(<\tau)}$ given $\mathcal{Q}_p^{(\tau)}$.

**Prototype-Guided Retrieval.** Given a query segment $\mathcal{Q}_p^{(\tau)}$, `EHR-RAGp` retrieves relevant historical segments from $\mathcal{H}_p^{(<\tau)}$ via a two-stage process: (1) coarse semantic retrieval to obtain candidate chunks, followed by (2) prototype-guided weighting to identify the most contextually aligned chunks based on agreement in the learned prototype space.

**Candidate Retrieval.** Both query chunk $\mathcal{Q}_p^{(\tau)}$ and historical chunks $h_i \in \mathcal{H}_p^{(<\tau)}$ are encoded using a frozen pretrained retriever $R_\beta$:

$$q' = R_\beta(\mathcal{Q}_p^{(\tau)}) \in \mathbb{R}^d, \qquad c_i' = R_\beta(h_i) \in \mathbb{R}^d.$$

Candidate chunks are retrieved by computing cosine similarity between $q'$ embedding and $\{c_i'\}$ stored in an index $\mathcal{I}$, selecting the top-$M$ most similar:

$$\{h_1, h_2, \ldots, h_M\} \subset \mathcal{H}_p^{(<\tau)}.$$

**Query and Candidate Encoding.** After retrieval, the query segment $\mathcal{Q}_p^{(\tau)}$ and the top-$M$ chunks $\{h_1, h_2, \ldots, h_M\}$ are encoded using a shared backbone $f_\alpha$ to obtain $q$ and $\{c_i\}$:

$$q = f_\alpha(\mathcal{Q}_p^{(\tau)}) \in \mathbb{R}^d, \qquad c_i = f_\alpha(h_i) \in \mathbb{R}^d.$$

Sharing $f_\alpha$ ensures a common embedding space for queries and chunks, enabling efficient semantic alignment and filtering of relevant context.

**Prototype-Guided Alignment.** `EHR-RAGp` maintains a learnable set of $L$ prototype vectors that organize longitudinal history into latent semantic modes:

$$P = \{p_1, p_2, \ldots, p_L\}, \qquad p_l \in \mathbb{R}^d,$$

These prototypes capture latent clinical patterns and are learned end-to-end, enabling automatic discovery of structure and alignment with downstream tasks.

For a chunk embedding $x \in \mathbb{R}^d$, the model computes a distribution over prototypes:

$$\pi(x) = \mathrm{softmax}\left(\frac{xP^\top}{T_\pi}\right) \in \mathbb{R}^L,$$

where $x$ corresponds to either the **query** $q$ or a **candidate chunk** $c_i$, and $T_\pi$ controls distribution sharpness (Assran et al., 2022). We use different temperatures for query and history, denoted $T_q$ and $T_h$ with $T_q < T_h$, reflecting the need for sharper query focus and broader history coverage.

Thus,

$$\pi_q = \pi(q), \qquad \pi_i = \pi(c_i), \ i = 1, \ldots, M.$$

Here, $\pi_q$ captures query–prototype alignment, while $\pi_i$ represents each chunk's association with the prototypes.

To measure alignment in the prototype space, `EHR-RAGp` computes the agreement between the query and candidate distributions using a cross-entropy-based score as $\alpha_i = -\sum_{l=1}^{L} \pi_q^{(l)} \log \pi_i^{(l)}$, where $\alpha \in [0, \infty)$. Lower values indicate stronger alignment, as $\pi_i$ better matches the prototype emphasis of $\pi_q$, while higher values indicate weaker alignment. This formulation enables smooth, differentiable matching without requiring hard prototype assignments.

**Prototype-Guided Weighting.** Given the prototype alignment scores $\{\alpha_i\}_{i=1}^{M}$, `EHR-RAGp` converts them into normalized importance weights using a temperature-controlled softmax:

$$w_i = \mathrm{softmax}\left(\frac{-\alpha_i}{T_s}\right),$$

where $T_s$ controls the smoothness of the weights.

`EHR-RAGp` uses a fully differentiable soft weighting mechanism, where all retrieved chunks contribute with varying

*Table 1.* Performance comparison between `EHR-RAGp` and clinical baselines.

| Model | ICU-Readmit 30d | | In-hospital Mortality | | Long LOS 7d | | 1YR Mortality | |
|---|---|---|---|---|---|---|---|---|
| | AUROC (CI) | AUPRC (CI) | AUROC (CI) | AUPRC (CI) | AUROC (CI) | AUPRC (CI) | AUROC (CI) | AUPRC (CI) |
| DescEmb | 0.660 (0.634, 0.684) | 0.096 (0.080, 0.118) | 0.933 (0.927, 0.940) | 0.693 (0.669, 0.716) | 0.849 (0.840, 0.858) | 0.480 (0.455, 0.505) | 0.766 (0.755, 0.777) | 0.329 (0.308, 0.351) |
| DescEmb*‡ | 0.632 (0.605, 0.658) | 0.098 (0.080, 0.124) | 0.911 (0.903, 0.918) | 0.631 (0.606, 0.655) | 0.815 (0.804, 0.825) | 0.403 (0.378, 0.427) | 0.740 (0.728, 0.752) | 0.304 (0.283, 0.327) |
| GenHPF | 0.705 (0.681, 0.728) | 0.134 (0.108, 0.167) | 0.932 (0.926, 0.939) | 0.706 (0.682, 0.730) | 0.850 (0.841, 0.859) | 0.482 (0.456, 0.510) | 0.781 (0.770, 0.793) | 0.345 (0.324, 0.369) |
| Med-BERT | 0.715 (0.693, 0.739) | 0.106 (0.091, 0.125) | 0.907 (0.899, 0.915) | 0.609 (0.581, 0.635) | 0.802 (0.790, 0.814) | 0.387 (0.364, 0.411) | 0.749 (0.738, 0.76) | 0.305 (0.285, 0.327) |
| CEHR-BERT | 0.718 (0.694, 0.739) | 0.133 (0.110, 0.163) | 0.933 (0.927, 0.940) | 0.700 (0.677, 0.724) | 0.838 (0.829, 0.848) | 0.447 (0.423, 0.472) | 0.778 (0.767, 0.789) | 0.342 (0.321, 0.365) |
| BEHRT | 0.737 (0.714, 0.761) | 0.133 (0.112, 0.162) | 0.925 (0.918, 0.932) | 0.674 (0.648, 0.697) | 0.831 (0.821, 0.841) | 0.439 (0.417, 0.463) | 0.771 (0.760, 0.783) | 0.337 (0.315, 0.361) |
| Hi-BEHRT | 0.596 (0.571, 0.621) | 0.055 (0.048, 0.067) | 0.897 (0.888, 0.904) | 0.585 (0.557, 0.611) | 0.775 (0.761, 0.787) | 0.355 (0.332, 0.380) | 0.700 (0.688, 0.712) | 0.254 (0.237, 0.273) |
| EHRMamba | 0.725 (0.702, 0.749) | 0.130 (0.109, 0.160) | 0.925 (0.918, 0.932) | 0.640 (0.614, 0.670) | 0.865 (0.855, 0.874) | 0.578 (0.552, 0.605) | 0.757 (0.746, 0.769) | 0.303 (0.283, 0.325) |
| REMed | 0.535 (0.510, 0.561) | 0.044 (0.039, 0.052) | 0.867 (0.857, 0.876) | 0.468 (0.439, 0.499) | 0.800 (0.790, 0.811) | 0.363 (0.342, 0.386) | 0.622 (0.608, 0.636) | 0.185 (0.175, 0.199) |
| EHR-RAGp (Ours) | **0.747 (0.724, 0.768)** | **0.156 (0.128, 0.189)** | **0.940 (0.933, 0.945)** | **0.716 (0.693, 0.738)** | **0.884 (0.875, 0.893)** | **0.618 (0.593, 0.644)** | **0.820 (0.811, 0.829)** | **0.393 (0.369, 0.418)** |

‡ DescEmb represents DescEmb BERT-FT variant, whereas DescEmb* represents DescEmb CLS-FT variant. For more information, see Appendix D.

*Table 2.* Performance gains of existing EHR baseline with `EHR-RAGp`.

| Model | EHR-RAGp | ICU-Readmit 30d | | In-hospital Mortality | | Long LOS 7d | | 1YR Mortality | |
|---|---|---|---|---|---|---|---|---|---|
| | | AUROC (CI) | AUPRC (CI) | AUROC (CI) | AUPRC (CI) | AUROC (CI) | AUPRC (CI) | AUROC (CI) | AUPRC (CI) |
| Med-BERT | × | 0.715 (0.693, 0.739) | 0.106 (0.091, 0.125) | 0.907 (0.899, 0.915) | 0.609 (0.581, 0.635) | 0.802 (0.790, 0.814) | 0.387 (0.364, 0.411) | 0.749 (0.738, 0.760) | 0.305 (0.285, 0.327) |
| | ✓ | **0.725 (0.703, 0.748)** | **0.121 (0.103, 0.147)** | **0.929 (0.922, 0.936)** | **0.683 (0.658, 0.709)** | **0.861 (0.851, 0.870)** | **0.586 (0.561, 0.610)** | **0.773 (0.761, 0.784)** | **0.307 (0.289, 0.328)** |
| CEHR-BERT | × | 0.718 (0.694, 0.739) | 0.133 (0.110, 0.163) | 0.933 (0.927, 0.940) | 0.700 (0.677, 0.724) | 0.838 (0.829, 0.848) | 0.447 (0.423, 0.472) | 0.778 (0.767, 0.789) | 0.342 (0.321, 0.365) |
| | ✓ | **0.745 (0.724, 0.767)** | **0.167 (0.137, 0.203)** | **0.939 (0.933, 0.946)** | **0.717 (0.694, 0.741)** | **0.876 (0.867, 0.885)** | **0.604 (0.580, 0.627)** | **0.802 (0.791, 0.811)** | **0.372 (0.350, 0.396)** |
| BEHRT | × | 0.737 (0.714, 0.761) | 0.133 (0.112, 0.162) | 0.925 (0.918, 0.932) | 0.674 (0.648, 0.697) | 0.831 (0.821, 0.841) | 0.439 (0.417, 0.463) | 0.771 (0.760, 0.783) | 0.337 (0.315, 0.361) |
| | ✓ | **0.743 (0.722, 0.765)** | **0.135 (0.112, 0.166)** | **0.932 (0.925, 0.939)** | **0.680 (0.656, 0.704)** | **0.871 (0.862, 0.879)** | **0.599 (0.575, 0.623)** | **0.792 (0.783, 0.803)** | **0.361 (0.338, 0.385)** |

importance. This preserves distributed clinical evidence across multiple chunks, which is crucial for EHR data.

The weights $\{w_i\}$ modulate each chunk's contribution during fusion, emphasizing relevant context while retaining complementary information.

**Retrieval-Augmented Fusion.** The query $q$ and retrieved chunks $\mathcal{R}_p^{(<\tau)} = \{c_1, \ldots, c_n\}$ are combined using prototype-guided weighting, where each chunk is scaled by $w_i$:

$$z_{\text{fused}} = \left(q, \; w_1 c_1, \; w_2 c_2, \; \ldots, \; w_n c_n\right).$$

This sequence is processed by a transformer encoder $g(\cdot)$ followed by a a linear classifier $h(\cdot)$ to yield the final prediction:

$$\hat{y}_p^{(\tau)} = h\left(g(z_{\text{fused}})\right).$$

**Training Objective.** `EHR-RAGp` is trained end-to-end using binary cross-entropy for the prediction task, while keeping the retriever $R_\beta$ frozen and updating $f_\alpha$, the prototypes, and the fusion module. The objective combines task loss with prototype usage regularization over query and history:

$$\mathcal{L} = l\left(y_p^{(\tau)}, \hat{y}_p^{(\tau)}\right) - \lambda_{\text{u}}\left(\mathcal{H}(\bar{\pi}_q) + \mathcal{H}(\bar{\pi}_h)\right),$$

where $\bar{\pi}_q = \frac{1}{B}\sum_{b=1}^{B}\pi_q^{(b)}$ and $\bar{\pi}_h = \frac{1}{\sum_b K_b}\sum_{b,k}\pi_{h_k}^{(b)}$ denote the average prototype assignment distributions over a batch for the query and historical chunks, respectively, $\mathcal{H}(\cdot)$ is the entropy function, and $\lambda_{\text{u}} > 0$ determines the strength of the regularization term. This regularization encourages the model to avoid degenerate solutions in which only a small subset of prototypes are used.

## 3. Experiments

**Dataset & Tasks.** We evaluate `EHR-RAGp` on the large-scale real-world MIMIC-IV dataset (v3.1) (Johnson et al.,

2020), which contains heterogeneous EHR data from 364,627 patients across 546,028 hospital admissions and 94,458 ICU stays (Johnson et al., 2023). The dataset includes two modules: *hosp*, covering general clinical data such as labs and medications, and *icu*, containing ICU-specific data such as chart events and infusions. We include all clinically relevant tables and exclude operational data (e.g., provider information). Further details are provided in Appendix B.

We evaluate `EHR-RAGp` on four risk prediction tasks: (1) in-hospital mortality, (2) ICU readmission within 30 days, (3) long length of stay (7 days or more), and (4) 1-year mortality post discharge. These tasks are standard benchmarks in EHR modeling, assessing the model's ability to capture acute risk, longitudinal progression, and healthcare utilization from structured patient histories. Further details are provided in Appendix C.2.

**Implementation Details.** We use RoFormer *base* (Su et al., 2024) as the backbone encoder. First, the encoder is pretrained using an MLM objective following prior work (Liu et al., 2019), and all code-based baselines are pretrained under the same setting for fair comparison. The pretrained encoder is then used to construct and query the vector database. In the second stage, the full `EHR-RAGp` model is fine-tuned end-to-end for downstream tasks using binary cross-entropy loss (Appendix C.2). Hyperparameters are tuned via Bayesian optimization (Appendix C). Models are evaluated using AUROC and AUPRC with 95% confidence intervals computed via percentile bootstrap (Puth et al., 2015).

**Baselines.** We compare `EHR-RAGp` to clinical EHR foundation models. Clinical baselines include EHR-specific foundation models such as DescEmb (Hur et al., 2022a), GenHPF (Hur et al., 2023), Med-BERT (Rasmy et al., 2021),

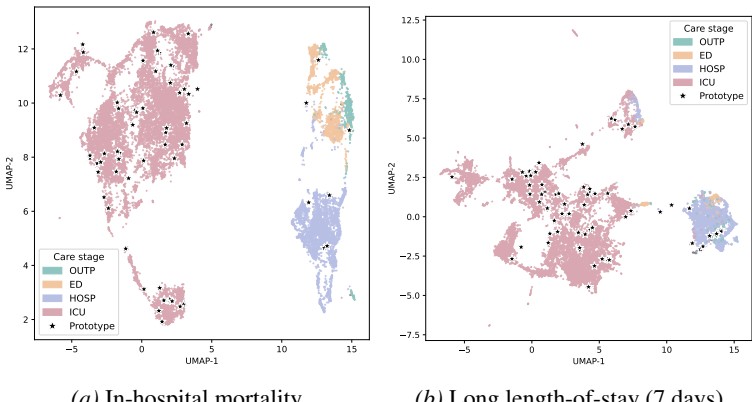

*(a)* In-hospital mortality     *(b)* Long length-of-stay (7 days)

*Figure 2.* UMAP projections of history embeddings and prototypes for (a) long length of stay and (b) in-hospital mortality tasks.

CEHR-BERT (Pang et al., 2021), BEHRT (Li et al., 2020), Hi-BEHRT (Li et al., 2022), EHRMamba (Fallahpour et al., 2024), and REMed (Kim et al., 2023). To ensure fair comparison, all models are adapted to a shared vocabulary and embedding layer, unifying the feature space across models.

## 4. Results

We present the main results of `EHR-RAGp` in this section, additional results are provided in Section E

**Overall Performance Results.** Table 1 reports AUROC and AUPRC across all baselines. `EHR-RAGp` achieves the best performance across all tasks. For ICU readmission, it reaches an AUROC of 0.747 (0.724, 0.768) and AUPRC of 0.156 (0.128, 0.189), outperforming BEHRT and CEHR-BERT. For in-hospital mortality, it attains an AUROC of 0.940 (0.933, 0.945) and AUPRC of 0.716 (0.693, 0.738), improving over strong baselines such as CEHR-BERT and GenHPF. For long length of stay, it achieves 0.884 (0.875, 0.893) AUROC and 0.618 (0.593, 0.644) AUPRC, surpassing EHRMamba and GenHPF. For 1-year mortality, it attains the top scores with AUROC 0.820 (0.811, 0.829) and AUPRC 0.393 (0.369, 0.418). Compared to REMed, `EHR-RAGp` shows substantial gains across all tasks, highlighting the benefit of retrieving diverse clinical event chunks. Overall, the results demonstrate consistent improvements, especially in the more challenging ICU readmission task.

**EHR-RAGp Improves Clinical Foundation Models.** Table 2 evaluates whether `EHR-RAGp` can improve existing EHR foundation models when used as a retrieval-augmented extension. Across Med-BERT, CEHR-BERT, and BEHRT, adding `EHR-RAGp` consistently improves performance across all four tasks. For Med-BERT, the largest gains appear in long LOS prediction, where AUPRC increases from 0.387 to 0.586, and in-hospital mortality, where AUPRC improves from 0.609 to 0.683. For CEHR-BERT, `EHR-RAGp` improves ICU readmission AUPRC from 0.133 to 0.167, long LOS AUPRC from 0.447 to

0.604, and 1-year mortality AUPRC from 0.342 to 0.372. Similar gains are observed for BEHRT, especially for long LOS, where AUPRC increases from 0.439 to 0.599. These results show that `EHR-RAGp` is not only a standalone model, but also a general retrieval-augmentation framework that can strengthen existing EHR foundation models.

**Latent Representation Structure.** Figure 2 shows UMAP projections of history embeddings for in-hospital mortality and long length-of-stay tasks. Across both tasks, the embedding space exhibits a consistent global structure, with clear separation between high-acuity (ICU) and lower-acuity (HOSP, ED, OUTP) regions, indicating organization primarily by clinical severity. Task-specific differences in cluster geometry further suggest adaptation to the prediction objective. Prototype centers (black stars) are distributed across dense and peripheral regions of the manifold, supporting their role in capturing diverse patient trajectory patterns. The visualizations demonstrate that the model learns structured, task-sensitive representations while preserving clinically meaningful care-stage organization. Figure S4 presents corresponding UMAP plots for ICU-Readmit 30d and 1YR Mortality. Additional qualitative analysis in Appendix E.3 examines prototype usage, chunk relevance weighting, and care-stage contributions during inference.

## 5. Conclusion

In this work, we introduce `EHR-RAGp`, a retrieval-augmented, prototype-guided foundation model for reasoning over heterogeneous EHR trajectories. By framing longitudinal modeling as a retrieval problem, `EHR-RAGp` overcomes fixed-window limitations and enables scalable integration of long-range context. Across multiple tasks, it consistently outperforms strong baselines, while prototype-guided retrieval improves efficiency by reducing training cost and context size. These results support the claim that expanding medical context improves predictive performance (Wornow et al., 2024; Fallahpour et al., 2024; Li et al., 2022).

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

## A. Related Work

**EHR Foundation Models:** Traditional EHR modeling approaches process patient trajectories as sequences of discrete clinical events collected over time. Some models primarily consider the occurrence of certain events, while others process the occurrence of the clinical event and its measured value, i.e. heart rate is 56 beats per minute. Early techniques utilized recurrent models such as Long-Short Term Memory (LSTM) (Hochreiter & Schmidhuber, 1997) or Gated Recurrent Unit (GRU) networks (Cho et al., 2014) to capture long-term dependencies (Che et al., 2018; Harutyunyan et al., 2019; Duan et al., 2019; Hayat et al., 2022).

The emergence of transformer-based language models, such as BERT (Devlin et al., 2019) and GPT-2 (Radford et al., 2019), paved the way for more advanced EHR foundation models. For example, encoder-based transformer models such as BEHRT (Li et al., 2020), Hi-BEHRT (Li et al., 2022), CEHR-BERT (Pang et al., 2021), and Med-BERT (Rasmy et al., 2021) introduced large-scale pretraining over longitudinal sequences of medical events such as diagnoses, procedures, labs, and medications, to enhance the contextualization of patient trajectories across visits. Other decoder-based transformer models such as ETHOS (Renc et al., 2025), CEHR-GPT (Pang et al., 2024), Curiosity (Waxler et al., 2025), and EHR2Path (Pellegrini et al., 2025) focused on the development of generative EHR foundation models for clinical prediction tasks. The common link between all these models is that they mimic standard language models by learning contextualized medical code vocabularies as tokens.

Another category of EHR foundation models shifts away from code-based modeling towards text-based modeling by treating each medical code as a textual description. Hence, the patient timeline constitutes a set of textual descriptions that can be processed directly by large language models (Hur et al., 2022b; 2023; Lee et al., 2025). Despite the remarkable progress, these models continue to inherit core limitations from their language-model roots. More specifically, they operate on truncated encounter windows, compress long and irregular patient histories into fixed-length sequences, and rely on restricted types of clinical events. While some methods have explored the use of longer context windows to accommodate extended patient histories (Waxler et al., 2025; Hur et al., 2022b; 2023), their vocabularies are heavily tokenized for medical codes and numerical values, or textualized and dependent on the base language models' reasoning capabilities. As a result, substantial portions of a patient's longitudinal trajectory remain inaccessible in downstream prediction tasks. This emphasizes the need for scalable frameworks that are capable of integrating and reasoning over complete EHR trajectories.

**Retrieval-Augmented Modeling:** RAG has emerged as a powerful technique in NLP for extending large language models capabilities by allowing them to condition on external, non-parametric knowledge sources. The original RAG framework (Lewis et al., 2020) demonstrated how combining dense retrieval with generative transformers improves factual correctness and reduces hallucinations. The concept was later extended by models such as RETRO (Borgeaud et al., 2022), FLARE (Jiang et al., 2023), and REPLUG (Shi et al., 2024). These architectures share a common principle: retrieval allows the model to incorporate long-range or domain-specific information without encoding the entire knowledge base within its internal parameters. The success of the retrieval paradigm has also influenced healthcare applications, particularly in tasks involving clinical question answering (Saba et al., 2024; Sohn et al., 2025), evidence grounding (Lewis et al., 2025; Jia et al., 2025), biomedical knowledge integration (Feng et al., 2025b;a), phenotyping and cohort identification (Ziletti & D'Ambrosi, 2025), and many other applications related to medical textual data (Abo El-Enen et al., 2025; Amugongo et al., 2025).

In the context of EHR modeling, RAG has been primarily explored as a mechanism for integrating external knowledge into predictive models, such as EMERGE (Zhu et al., 2024a), RAM-EHR (Xu et al., 2024), REALM (Zhu et al., 2024b), and KAMELEON (Datta et al., 2025). To the best of our knowledge, REMED (Kim et al., 2023) is the most closely related work to our proposed approach. REMED's core idea is to retrieve the most relevant subset of medical codes for a given outcome prediction task. Their goal is to overcome manual selection of clinical events or features. Overall, recent advancements highlight the potential of retrieval-augmented modeling for EHR, but they also reveal a clear unmet need for mechanisms that can dynamically access and integrate a patient's entire structured history.

## B. Data Preprocessing

### B.1. Cohort construction

**Pretraining Cohort:** We first convert the MIMIC-IV dataset into the Medical Event Data Standard (MEDS) format (Arnrich et al., 2024; McDermott et al., 2025), representing each patient history as a chronologically ordered sequence of medical events covering all patient visits within a single sequence. We then filter the dataset to construct the pretraining

cohort. Specifically, we exclude patients listed in the patients table who do not have any valid hospital or ICU admissions. Next, we examine the total number of events within each visit and exclude visits containing fewer than 10 events. At this stage, we do not apply any age-based filtering, as this cohort is intended solely for pretraining. We further exclude all patients included in the test set from the pretraining cohort to prevent data leakage during evaluation. The resulting cohort consists of 199,012 patients with 463,436 hospital admissions, 80,005 ICU stays, and 496,199,673 medical events.

**Downstream Cohort:** For downstream tasks, we construct the dataset using ICU stays only and exclude patients with hospital admissions but no ICU stays. We follow the inclusion and exclusion criteria proposed by (Harutyunyan et al., 2019) to define the downstream cohort. First, we include all adult patients with an age at admission of $\geq 18$ years. Second, we retain patients with a single ICU stay within the same hospital admission. Third, we include only patients whose admission and discharge ICU units are identical. Finally, we examine the length of each valid ICU stay and retain only those stays lasting at least 24 hours. This process results in a downstream cohort comprising 49,839 patients with 61,175 ICU stays.

### B.2. Data Preprocessing

We perform comprehensive preprocessing to reduce irregularities and transform the data into a standardized format. First, MIMIC-IV contains ICD diagnosis and procedure codes from both versions 9 and 10. To unify all codes under a single coding system, we map all ICD-9 diagnosis and procedure codes to ICD-10. We use the General Equivalence Mappings (GEMS[1]) released in 2018, which is the most recent version available.

Second, we examine medication events and remove any medications appearing in the patient timeline with ambiguous or unclear names. For laboratory tests, we first identify all lab entries listed in the MIMIC-IV file *d_labitems_to_loinc.csv* that are marked as non-laboratory measurements and filter them out. Subsequently, we modify lab event naming to incorporate the lab *item_id*, specimen fluid, and test name. This naming convention preserves the original structure of the MIMIC-IV dataset, increases the expressiveness of laboratory events, and reduces semantic ambiguity.

For ICU data, we modify event names to explicitly reflect the source table. For example, events originating from the *chartevents.csv* table are prefixed with *ICU-CHART*. We further include the corresponding *item_id* as a secondary identifier, followed by the event-specific name. Representative examples are provided in Table S1.

Next, we handle outliers in numeric values using the *MEDS-transform* Python package by computing dataset-wide statistics for each numerical event and excluding values that lie beyond three standard deviations from the mean. Finally, we normalize all remaining numeric values using *MEDS-transform* by recomputing normalization statistics after outlier removal.

### B.3. Patient Timeline Construction

Figure S1 depicts a sample patient timeline segmented into distinct care stages and covering all relevant clinical information. To construct a patient timeline, we first examine each patient sequence and identify hospital admissions (HOSP) using the *hadm_id* field, as well as ICU stays (ICU) using the *icustay_id* field, which are the only care stages in MIMIC-IV associated with unique identifiers.

During this process, we observe that certain medical events, such as laboratory tests and microbiology orders, appear in the patient timeline without an explicit reference to any hospital admission. To handle these events without excluding them, we assess their temporal proximity to admissions present in the patient timeline. Based on the time difference, we associate each such event with the nearest hospital admission by assigning the corresponding *hadm_id*. Events occurring within 24 hours of the nearest hospital admission are labeled as emergency department (ED) events, while events occurring within 30 days are treated as outpatient (OUTP) events.

To represent time gaps between consecutive visits, we insert artificial time tokens following prior work (Pang et al., 2021; Fallahpour et al., 2024), denoted as GAP, as illustrated in Figure S1. These tokens encode temporal intervals spanning weeks, months, or years, depending on the elapsed time between visits.

A standard patient timeline primarily consists of hospital events including laboratory measurements, medications, microbiology tests, ICD procedure codes, ICD diagnosis codes, and DRG codes, as well as ICU events such as chart events, infusions, procedures, and fluid outputs. In addition, administrative events are included for both hospital and ICU stays to indicate boundaries, locations, and care types. To explicitly preserve care-stage boundaries within the timeline, we introduce special

---

[1]https://www.cms.gov/medicare/coding-billing/icd-10-codes/icd-10-cm-icd-10-pcs-gem-archive

boundary tokens, including OUTPATIENT-START, OUTPATIENT-END, EMERGENCY-START, and EMERGENCY-END for OUTP and ED stages, respectively.

Demographic attributes such as gender and race are treated as static events without timestamps and are prepended to the beginning of each sequence. Patient age is computed at the start of each stay and represented as a dedicated event token, AGE-AT-ADMISSION, with an associated numeric value. If a patient dies during or after discharge, a special timestamped token, MEDS_DEATH, is appended to the sequence. We also include special tokens required for language modeling objectives, such as PAD, MASK, CLS, and UNK, where applicable. Table S1 summarizes the different event types present in the patient timeline along with their frequencies.

For events associated with numeric values, such as laboratory tests and ICU chart events, we attach the corresponding numeric measurements. For events without numeric values, including PROCEDURE-ICD and DIAGNOSIS-ICD, we associate a learnable *null* parameter. To encode temporal information, we compute the time difference between consecutive events in minutes and attach this value to each event. Because time gaps may vary substantially both within and across visits, we scale time deltas using the transformation defined in Equation 1, which maps values to the range $[0, 1]$ and prevents extreme magnitudes. Temporal representations are then processed using Time2Vec (Kazemi et al., 2019) during training.

$$\Delta t_i^{'} = \frac{log(1 + \Delta t_i)}{log(\Delta t_{max})} \tag{1}$$

where $\Delta t_i$ is the consecutive time difference at event $i$, and $\Delta_{max}$ is the maximum time difference present in the dataset.

To further contextualize each event within the patient timeline, we introduce three additional categorical representations, care stage, visit order, and event type, as illustrated in the bottom rows of Figure S1. The care stage embedding explicitly encodes the clinical context in which each event occurs, distinguishing between outpatient (OUTP), emergency department (ED), inpatient hospitalization (HOSP), ICU stay (ICU), and artificial gap (GAP) events. This representation enables the model to differentiate identical medical events occurring under different clinical settings, which often carry distinct semantic meanings.

In addition, we assign a visit order index to each event, indicating the chronological visit number to which it belongs. The visit order is incremented across care episodes and reset for time gap tokens, allowing the model to reason over longitudinal disease progression across multiple encounters rather than treating the timeline as a flat sequence. Finally, the event type embedding encodes the high-level category of each event, such as laboratory tests, medications, diagnoses, procedures, administrative markers, or special tokens. Below, we provide a formal definition of each component in that define a complete event:

1. **Concept Embedding:** Each clinical concept $c_{nk}$ (e.g., diagnosis, lab test, procedure, medication) is associated with a learnable semantic embedding: $\mathbf{u}_{c_{nk}} \in \mathbb{R}^d$.

2. **Value Embedding:** For events with a numerical measurement $v_{nk}$, the value is normalized and projected, such that $\mathbf{v}_{nk} = \theta(v_{nk}) \in \mathbb{R}^d$, and $\theta(\cdot)$ is a multi-layer perceptron. Events without values are assigned a learnable (null-value) parameter.

3. **Time Encoding:** Each timestamp $t_{nk}$ is mapped into a temporal embedding representing the local temporal information as time deltas between consecutive events, such that $\mathbf{t}_{nk} \in \mathbb{R}^d$.

4. **Visit Order Embedding:** To capture patient-level longitudinal ordering, each event inherits a visit index embedding: $\mathbf{r}_{\text{visit}(n)} \in \mathbb{R}^d$, where the indices are encoded as continuous learnable positional vectors. This allows the model to distinguish between early and late episodes in the patient trajectory (global temporal representation).

5. **Care Stage Embedding:** Each visit belongs to a clinical stage (e.g., outpatient, emergency, inpatient, ICU). We assign a learned embedding: $\mathbf{s}_{\text{stage}(n)} \in \mathbb{R}^d$ to enable the model to incorporate care-context information, e.g., ICU vs. emergency events carry different clinical semantics.

6. **Type Embedding:** Each event also receives a higher-level event type identifier that encodes the data source from which it originates (ICU chart event, ICU fluid output, etc.): $\mathbf{w}_{\text{type}(nk)} \in \mathbb{R}^d$. This provides local structural information that allows the model to differentiate heterogeneous event types.

| | Visit 1 | | | | | | | | | | | | | Visit 2 | | | | | | |
|---|---|---|---|---|---|---|---|---|---|---|---|---|---|---|---|---|---|---|---|---|
| | OUTP | | GAP | ED | | | HOSP | | | | GAP | ED | | HOSP | | | ICU | | | HOSP | |
| Concept | $L_1$ | $L_2$ | $T$ | $L_1$ | $L_2$ | $L_3$ | $P_1$ | $P_2$ | $D_1$ | $D_2$ | $T$ | $L_1$ | $L_2$ | $P_1$ | $M_1$ | $M_2$ | $P_2$ | $M_3$ | $M_4$ | $D_1$ | $D_2$ |
| Value | $x_l$ | $x_l$ | null | $x_l$ | $x_l$ | $x_l$ | null | null | null | null | null | $x_l$ | $x_l$ | null | null | null | null | null | null | null | null |
| Time | $\Delta t$ | $\Delta t$ | $\Delta t$ | $\Delta t$ | $\Delta t$ | $\Delta t$ | $\Delta t$ | $\Delta t$ | $\Delta t$ | $\Delta t$ | $\Delta t$ | $\Delta t$ | $\Delta t$ | $\Delta t$ | $\Delta t$ | $\Delta t$ | $\Delta t$ | $\Delta t$ | $\Delta t$ | $\Delta t$ | $\Delta t$ |
| Stage | 1 | 1 | 0 | 2 | 2 | 3 | 3 | 3 | 3 | 3 | 0 | 2 | 2 | 3 | 3 | 3 | 4 | 4 | 4 | 3 | 3 |
| Visit | 1 | 1 | 0 | 1 | 1 | 1 | 1 | 1 | 1 | 1 | 0 | 2 | 2 | 2 | 2 | 2 | 2 | 2 | 2 | 2 | 2 |
| Type | 1 | 1 | 2 | 1 | 1 | 1 | 3 | 3 | 4 | 4 | 2 | 1 | 1 | 3 | 5 | 5 | 3 | 5 | 5 | 4 | 4 |

*Figure S1.* Sample patient timeline consisting of two visits. We illustrate the main components constituting the patient timeline in our implementation including medical events, numeric values, time values, care stage representation, visit order representation, and event type representation. Abbreviations, OUTP: Outpatient, ED: Emergency Department, HOSP: Hospital Admission, ICU: Incentive Care Unit, L: Labs, T: TIME-GAP, P: Procedure, M Medication, D: Diagnosis, $x_a$ numeric value. This is a sample timeline created randomly for illustration purposes and does not reflect real data.

By jointly modeling care stage, visit order, and event type, the patient timeline representation preserves structural, temporal, and semantic distinctions across events, facilitating more expressive and context-aware representation learning over heterogeneous EHR sequences.

## B.4. Query/History Definition

For each prediction instance in the downstream cohort, we decompose the patient timeline into a *query* segment and a corresponding *history* segment to enable retrieval-based conditioning. The query represents the patient's current clinical context at the time of prediction and consists of events observed within a task-specific observation window. For details on the observation windows used for each task, we refer the reader to Section C.2 and Table S2. All events occurring strictly before the query window constitute the patient history and are segmented into fixed-length units that serve as retrievable candidates. Figure S2 illustrates the query/history definitions for the various prediction windows considered in this work. This formulation mirrors real-world clinical reasoning, in which decisions are made based on the present encounter while selectively consulting relevant past visits. By explicitly separating query and history, the model is encouraged to retrieve and integrate only the most informative historical context, rather than uniformly encoding the entire past, enabling scalable reasoning over long and heterogeneous patient trajectories.

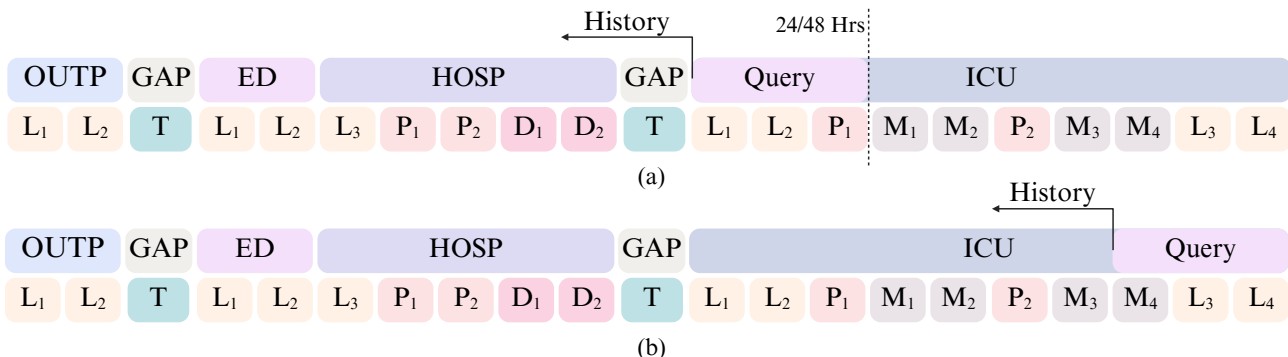

*Figure S2.* Illustration of query and history definition (a): Query/history boundaries for tasks with time-bounded prediction window. (b) Query/history boundaries task with entire stay prediction window.

## B.5. Chunking

Structured EHR trajectories are inherently irregular, with clinical events occurring at variable temporal resolutions and across diverse care settings. To organize longitudinal patient history into retrievable units, EHR-RAGp partitions the timeline into clinically coherent chunks. We investigate four chunking strategies: *event-based*, *time-based*, *visit-level*, and *care-stage* chunking. These strategies provide complementary views of patient history by capturing different temporal and clinical

*Table S1.* Summary of the various events present in the patient timeline. We summarize all events present in the patient timeline and list the count along with representative examples. The == sign indicates that event type and example are identical.

| Event type | # of unique events | Example |
|---|---|---|
| **Administrative Events** | | |
| OUTPATIENT-START | 1 | == |
| OUTPATIENT-END | 1 | == |
| EMERGENCY-START | 1 | == |
| EMERGENCY-END | 1 | == |
| ADMISSION-AT-HOSPITAL | 1 | == |
| ADMISSION-AT-ICU | 1 | == |
| ADMISSION-LOCATION | 12 | ADMISSION-LOCATION//TRANSFER FROM HOSPITAL |
| ADMISSION-TYPE | 9 | ADMISSION-TYPE//EW EMER |
| AGE_AT_ADMISSION | 1 | == |
| DISCHARGE-FROM-HOSPITAL | 1 | == |
| DISCHARGE-FROM-ICU | 1 | == |
| DISCHARGE-LOCATION | 14 | DISCHARGE-LOCATION//HOME |
| **Static Events** | | |
| Gender | 2 | GENDER//F |
| Race | 11 | RACE//HISPANIC |
| MEDS_DEATH | 1 | == |
| **Medical Events** | | |
| LAB | 851 | LAB//51237//Blood//INR(PT) |
| MEDICATION | 7486 | MEDICATION//heparin |
| MICROBIOLOGY | 169 | MICROBIOLOGY//90039//URINE CULTURE |
| PROCEDURE-ICD | 13711 | PROCEDURE-ICD//3E0G76Z |
| DIAGNOSIS-ICD | 21135 | DIAGNOSIS-ICD//K429 |
| DRG | 1087 | DRG//APR//228 |
| **ICU Events** | | |
| ICU-CHART | 2311 | ICU-CHART//220228//Hemoglobin |
| ICU-FLUID-OUTPUT | 71 | ICU-FLUID-OUTPUT//226559//Foley |
| ICU-PROCEDURE | 151 | ICU-PROCEDURE//225752//Arterial Line |
| ICU-INFUSION | 327 | Norepinephrine infusion |
| **Special Tokens** | | |
| PAD | 1 | == |
| MASK | 1 | == |
| CLS | 1 | == |
| UNK | 1 | == |
| **Time Tokens** | | |
| Week tokens: W1-W3 | 3 | TIME-GAP//1-W |
| Month tokens: M1-M12 | 12 | TIME-GAP//11-M |
| Year token: 1Y-Y+ | 1 | TIME-GAP//1-Y+ |

granularities. Figure S3 visually illustrates these chunking strategies. Below, we formally outline our proposed chunking strategies:

- **Event-Based:** Events are segmented into fixed-size groups based on event count. Each chunk contains a predefined number of embedded clinical events. Consecutive chunks overlap by a fixed number of events to preserve continuity across chunk boundaries.

- **Time-Based:** Events are segmented into fixed time windows, for example 6-hour, 12-hour, 24-hour blocks. Each chunk contains all events occurring in its temporal window.

- **Visit-level:** Each hospital visit, admission, or encounter is treated as a candidate chunk. This aligns with natural clinical boundaries: emergency department stays, inpatient admissions, and outpatient encounters.

- **Care-Stage:** Events are grouped by clinical stage (outpatient, emergency, inpatient, ICU), ensuring that each chunk reflects a coherent care stage.

In cases where the chunk length is shorter than the context window, we pad the sequence to match the context length. In

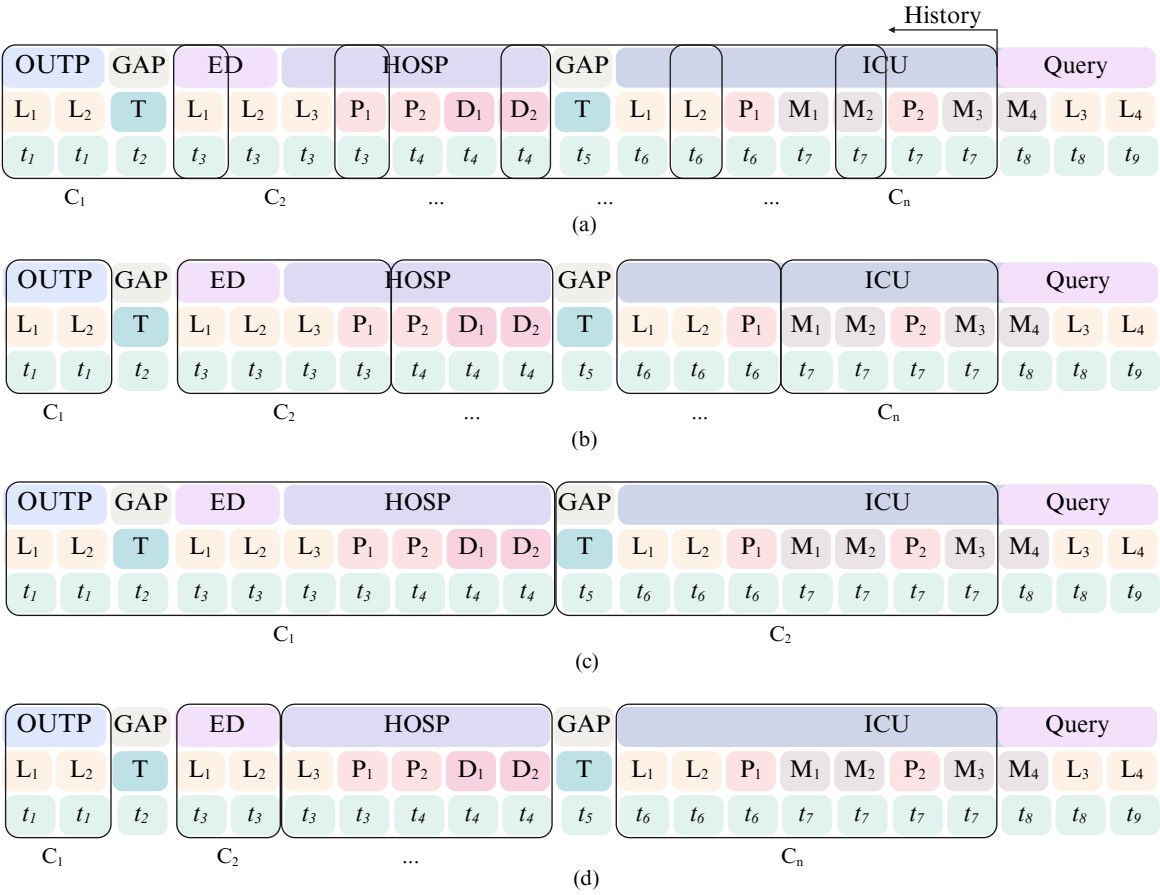

*Figure S3.* Visual illustration of the chunking strategies employed with `EHR-RAGp`. (a) Event-based chunking. (b) Time-based chunking. (c) Visit-level chunking. (d) Care-stage chunking.

cases where the chunk length is greater than the context length, we recursively impose *Event-Based Chunking* with proper metadata handling.

## C. Experimental Setup

### C.1. Data Splitting

To ensure a fair evaluation and prevent information leakage across training stages, we adopt a strict data splitting strategy. All data splits are performed at the *patient* level via *subject_id* field rather than the admission or visit level to ensure that no clinical information from the same patient appears in more than one split. For downstream prediction tasks, we partition the eligible ICU cohort into mutually exclusive training, validation, and test sets with proportions of $70\%$, $10\%$, and $20\%$, respectively. The test set is held out entirely and is not used during any stage of model development, including pretraining, hyperparameter tuning, or early stopping. This guarantees that the pretrained representations do not indirectly encode information from evaluation patients. Instead, the validation set is used exclusively for model selection and hyperparameter optimization. Further, for pretraining we hold out $5\%$ as validation set to examine the pretraining quality. This strict separation across patients and training stages ensures that reported performance accurately reflects the model's ability to generalize to unseen patients and supports reliable comparison across baselines.

### C.2. Downstream Tasks

- **In-Hospital Mortality:** This task aims to predict whether a patient will die during the current hospital admission as a binary classification problem, using information available up to a predefined observation window. It is a widely used benchmark for evaluating clinical risk prediction models. In our setup, we condition the in-hospital mortality task on

the first $48$ hours spent in the ICU, following (Harutyunyan et al., 2019; Hayat et al., 2022; Elsharief et al., 2025).

- **1-Year Mortality Post-Discharge:** This task focuses on predicting whether a patient will die within one year following hospital discharge or no. It captures longer-term outcomes and reflects the model's ability to leverage both acute and chronic patterns in a patient's longitudinal history. We consider the entire ICU stay data as a valid prediction window (Meng et al., 2024; Nistal-Nuño, 2022; Ghassemi et al., 2014). For patients with multiple ICU stay and positive label of mortality within 1-year, we only assign the positive label to all stays that are within one year from the mortality date.

- **Long Length of Stay:** This task predicts whether a patient's hospital stay will exceed a predefined duration threshold. It serves as a proxy for resource utilization and clinical complexity and is commonly used to assess a model's ability to identify patients at risk of prolonged hospitalization. We set the length-of-stay threshold to 7 days following (Wornow et al., 2024; 2023a; Kim et al., 2023) and define the prediction window as the first $24$ hours spent in the ICU, following (Zhang & Kuo, 2024; Alghatani et al., 2021).

- **ICU Readmission:** This task aims to predict whether a patient will be readmitted to the intensive care unit within a specified time window after ICU discharge. It is an important indicator of care quality and patient instability and requires effective modeling of prior clinical trajectories. We set the readmission window to 30 days post-discharge following (Hu et al., 2025; Wornow et al., 2023a; Chen et al., 2022) and use the entire ICU stay as the prediction window, following (Wornow et al., 2023a; de Sá et al., 2023).

*Table S2.* **Overview of the downstream tasks and the query/ history definition for each task**

| Task | Label type | Prediction Window | Reference point | Query Window | History Chunk |
|---|---|---|---|---|---|
| In-hospital mortality | Binary | 48 hours since ICU admission | ICU admission | Last $L$ event $\leq$ ICU admission $+$ 48 hours | All events $<$ query start |
| Long Length of stay-7 Days | Binary | 24 hours since ICU admission | ICU admission | Last $L$ event $\leq$ ICU admission $+$ 24 hours | All events $<$ query start |
| ICU Readmission-30 Days | Binary | Entire ICU stay | ICU discharge | Last $L$ event $\leq$ ICU discharge | All events $<$ query start |
| 1 Year Mortality | Binary | Entire ICU stay | ICU discharge | Last $L$ event $\leq$ ICU discharge | All events $<$ query start |

*Table S3.* **Label distribution per data split for downstream prediction tasks.**

| Task | Train (stay) | | Validation (stay) | | Test (stay) | |
|---|---|---|---|---|---|---|
| | N Pos (%) | N Neg (%) | N Pos (%) | N Neg (%) | N Pos (%) | Neg (%) |
| In-hospital mortality | $4,445$ (10.415%) | $38,234$ (89.585%) | $621$ (10.126%) | $5,512$ (89.874%) | $1,210$ (9.787%) | $11,153$ (90.213%) |
| Long Length of stay-7 Days | $5,810$(13.613%) | $36,869$(86.387%) | $835$ (13.615%) | $5,298$ (86.385%) | $1,637$ (13.241%) | $10,726$ (86.759%) |
| ICU Readmission-30 Days | $1,558$ (3.651%) | $41,121$ (96.349%) | $230$ (3.750%) | $5,903$ (96.250%) | $480$ (3.883%) | $11,883$ (96.117%) |
| 1 Year Mortality | $5,741$ (13.452%) | $36,938$ (86.548%) | $842$ (13.729%) | $5,291$ (86.271%) | $1,654$ (13.379%) | $10,709$ (86.621%) |

## C.3. `EHR-RAGp` Architecture

`EHR-RAGp` is built on a RoFormer-base (Su et al., 2024) encoder configured with 12 transformer layers, 12 attention heads, a hidden size of 768, and rotary positional embeddings applied at each self-attention layer, with a maximum positional embedding length of 1536. The encoder operates over fixed-length event sequences of $C = 1024$ tokens and produces contextualized token representations, from which sequence-level representations are obtained using either *CLS* pooling or mean pooling. We set the chunk overlap to $O = 12.5\%$ of the original chunk length.

Each event token is represented as the sum of multiple embedding components. In addition to the core event embedding, we incorporate categorical embeddings for care stage, visit order, and event type, all projected to the same hidden dimension (768) and learned jointly with the model. Numerical values associated with events are normalized and projected through a small MLP before being fused with the categorical embeddings. In the absence of a numeric value, we use a learnable null embedding of the same dimensionality. Temporal information is encoded using Time2Vec (Kazemi et al., 2019), applied to scaled time-delta features and added to the token representation.

For retrieval, the pretrained backbone encoder is used to embed historical timeline chunks, which are stored in a vector database. During training, the query representation is used to retrieve the top-$M$ most similar history chunks based on cosine similarity in the embedding space. Retrieval is non-parametric and does not involve gradient updates during the lookup stage; however, the encoder itself is fine-tuned end-to-end during downstream training.

The prototype module consists of a learnable set of task-specific prototype vectors defined in the same embedding space as the query and history representations. Both query and retrieved history embeddings are projected into the prototype space, where relevance scores are computed via similarity between history embeddings and prototypes, conditioned on the query representation. These scores are used to weight or filter retrieved history chunks, enabling selective integration of long-range context.

The fusion module combines the query representation with the filtered history representations using a shallow transformer with 2 layers and 4 attention heads. The fused representation is then passed to a task-specific prediction head implemented as a lightweight feedforward network. For all downstream tasks, the prediction head outputs a single logit optimized using binary cross-entropy (BCE) loss. All architectural components beyond retrieval are differentiable and optimized jointly during supervised fine-tuning.

### C.4. Pretraining Details

We adopt masked language modeling (MLM) objective applied to structured EHR event sequences for `EHR-RAGp` pretraining. During pretraining, patient timelines are tokenized into fixed-length sequences. We consider the full patient timeline during pretraining and pass it to the encoder backbone as overlapped chunks of 1024 tokens with 128 tokens of overlap. Hence, we produce dense sequences, whereas padding occurs only in the last chunk to fulfill the sequence length. During pretraining, a subset of 15% of the chunk events is randomly selected for masking following the RoBERTa (Liu et al., 2019) masking strategy. Specifically, masked positions are replaced with a special `[MASK]` token with probability 80%, substituted with a random event token with probability 10%, or left unchanged with probability 10%. The model is trained to predict the original event identity at masked positions using a cross-entropy loss over the entire vocabulary of 47,377 tokens.

All embedding components, including event embeddings and categorical embeddings, are trained jointly during this stage. No temporal, or numeric value components are used during pretraining; our objective at this stage is solely to learn contextualized representations of structured EHR sequences. Pretraining is conducted on the full pretraining cohort described in Section B.1.

We use *AdamW* (Loshchilov & Hutter, 2017) optimizer with an initial learning rate specified using cyclical learning rates following (Smith, 2017), a weight decay of $1 \times 10^{-2}$, a batch size of 16, and a cosine annealing learning scheduler. Training is performed for 100 epochs over the pretraining corpus. We monitor pretraining quality using accuracy on a held-out validation set, as described in Section C.1. Early stopping is implemented with a patience of three epochs if validation accuracy does not improve. The resulting pretrained backbone is subsequently reused for downstream fine-tuning and retrieval embedding generation. All downstream evaluation experiments are conducted using 4 NVIDIA A100 GPUs.

### C.5. Vector Database Setup

To enable retrieval-augmented conditioning, we construct a patient-specific vector database that stores embeddings of historical visit segments derived from the pretrained `EHR-RAGp` backbone. We use Facebook AI Similarity Search (FAISS[2]) as the vector index for `EHR-RAGp`. For each patient in the downstream cohort, the history portion of the timeline (as defined in Section B.4) is segmented into fixed-length chunks, where each chunk corresponds according ti the various chunking strategies proposed in our work. Each chunk is independently encoded using the pretrained backbone encoder, and the resulting pooled representation is stored as a dense vector.

We use mean pooling over the sequence representations and normalize the resulting embeddings prior to storage in the index. The vector index is constructed offline before downstream training and remains fixed during supervised fine-tuning. This design avoids repeated recomputation of historical embeddings and ensures scalable retrieval over long patient histories. During retrieval, we perform similarity search based on cosine similarity between the query embedding and historical chunk embeddings. It is worthy to note as well that we do not store chunked data along with their encoded vector, instead; we keep track of the chunk indices in the full chunked patient timeline. Upon retrieval, we perform online chunking during retrieve selected candidates by similarity search via their indices. This approach allow for faster retrieval and reduce the need for maintaining large database that accommodates the chunked data along with the vector index.

---

[2]https://github.com/facebookresearch/faiss

*Table S4.* Hyperparameter settings and search spaces used in `EHR-RAGp` experiments.

| Category | Hyperparameter | Fixed | Search Space |
|---|---|---|---|
| Backbone | Encoder architecture | RoFormer-base | – |
| | Hidden dimension ($d$) | 768 | – |
| | Number of layers | 12 | – |
| | Number of attention heads | 12 | – |
| | pooling | – | {mean, CLS} |
| Retrieval | Query Chunk size | 1024 | – |
| | History Chunk size | 256 | – |
| | History Chunk overlap* | 32 | – |
| | Number of retrieved chunks ($M$) | 24 | – |
| | Similarity metric | Cosine | – |
| Prototypes | Number of prototypes | – | {64, 128, 256, 512} |
| | Prototype dimension | 768 | – |
| | Prototype temperature ($T_q, T_h$) | – | {(0.025, 0.1), (0.05, 0.2), (0.02, 0.08)} |
| | Weighing temperature $T_s$ | – | {0.1, 0.15, 0.2, 0.25} |
| Fusion | Fusion layers | 2 | – |
| | Fusion attention heads | 4 | – |
| | pooling | – | {mean, query} |
| | Dropout | 0.1 | – |
| Training | Optimizer | SGD | – |
| | Usage penalty $\lambda_u$ | – | {0.004, 0.005, 0.006, 0.007} |
| | Learning rate | – | [1e−5, 5e−4] |
| | Weight decay | – | [1e−3, 1e−2] |
| | Batch size | – | {8,12,16,20} |
| | Max epochs | 75 | – |

* overlap is used only with event-based chunking strategy

## C.6. Downstream Training Details

Table S4 summarizes the hyperparameter settings and search spaces used for downstream training of `EHR-RAGp`. Core backbone parameters are fixed across all experiments, including the RoFormer-base encoder with a hidden dimension of 768, 12 layers, and 12 attention heads.

Retrieval-related hyperparameters including query chunk size, history chunk size, history chunk overlap, number of retrieved chunks, and similarity metric. all retrieval hyperparameters are held constant with values of 1024, 256, 32, 24, *cosine similarity*, respectively.

For prototypes module, we use fixed prototype dimensionality of 768. For temperature, we experiment with temperature pairs , $(T_q, T_h)$, for prototype assignment of $[(0.025, 0.1), (0.05, 0.2), (0.02, 0.08)]$ and Weighing temperature $T_s$ values of $\{0.1, 0.15, 0.2, 0.25\}$, whereas the number of prototypes is set to be optimized with discrete search space in $\{64\,128, 256, 512\}$.

For fusion, we set up a transformer encoder of 2 layers and 4 attention heads, and dropout of 0.1. We set the pooling strategy to be set during hyperparameter optimization with search space of $\{mean, query\}$.

For training hyperparameters, we experiment with Stochastic Gradient Descent (SGD) optimizer, learning rate in $[1e-5, 5e-4]$, Weight decay in $[1e-3, 1e-2]$ and cosine annealing learning rate scheduler. For usage penalty $\lambda_u$, we experiment with fixed values in $\{0.004, 0.005, 0.006, 0.007\}$. Batch size is set to be optimized with discrete search space of $\{8, 12, 16, 20\}$ and maximum epochs remain fixed with values of 75.

For hyperparameters optimization, we run 25 trails for each prediction task using Bayesian search. All downstream evaluation experiments were conducted using a single NVIDIA H100 NVL 94GB GPUs. Maximum batch size with cane be used with such a GPU in our experiments is 20 query chunks of length 1024 events along with 24 history chunk of length 256, while average epoch time given the max settings is 38 minutes. It is worthy to note that lower capacity GPUs can be also used with reduction in the max settings mentioned earlier.

## D. Baselines

### D.1. Clinical Baselines

We consider **encoder-based** clinical baselines, as they match the structure of `EHR-RAGp` as an encoder-based model.

- **DescEmb** (Hur et al., 2022a): is a text-based framework that represents patient history as textual descriptions of medical events, covering the event type, numeric value (if any), along with its unit of measurement. These sequences are then processed via clinical or general-purpose pretrained language models. The main goal of DescEmb is to reduce reliance on predefined medical code vocabularies through a shared linguistic vocabulary that can be utilized across different EHR datasets. We consider two variants of DescEmb in our experiments, including *BERT Fine-Tune* and *CLS*

*Table S5.* Impact of choice of chunking strategy on model performance across all tasks.

| Model | ICU-Readmit 30d | | In-hospital Mortality | | Long LOS 7d | | 1YR Mortality | |
|---|---|---|---|---|---|---|---|---|
| | AUROC (CI) | AUPRC (CI) | AUROC (CI) | AUPRC (CI) | AUROC (CI) | AUPRC (CI) | AUROC (CI) | AUPRC (CI) |
| Event-Based | **0.747** (0.724, 0.768) | **0.156** (0.128, 0.189) | 0.939 (0.933, 0.945) | 0.710 (0.684, 0.732) | 0.884 (0.875, 0.891) | 0.611 (0.586, 0.634) | 0.817 (0.807, 0.827) | 0.390 (0.365, 0.416) |
| Time-Based | 0.745 (0.722, 0.766) | 0.152 (0.126, 0.185) | 0.937 (0.930, 0.943) | 0.706 (0.683, 0.730) | 0.884 (0.876, 0.893) | 0.623 (0.598, 0.647) | 0.807 (0.797, 0.817) | 0.377 (0.353, 0.402) |
| Visit-Level | 0.738 (0.715, 0.762) | 0.153 (0.125, 0.184) | 0.939 (0.933, 0.945) | 0.715 (0.692, 0.737) | **0.885** (0.876, 0.893) | **0.628** (0.603, 0.652) | **0.821** (0.811, 0.831) | **0.396** (0.372, 0.422) |
| Care-Stage | 0.743 (0.720, 0.764) | 0.150 (0.124, 0.182) | **0.940** (0.933, 0.945) | **0.716** (0.693, 0.738) | 0.884 (0.875, 0.893) | 0.618 (0.590, 0.645) | 0.820 (0.811, 0.829) | 0.393 (0.369, 0.418) |

*Fine-Tune*.

- **GenHPF** (Hur et al., 2023): is an extension of DescEmb that goes beyond the basic description of the medical event type, value, and unit to include all features associated with an event in the raw EHR tables, enhancing medical event representation. For instance, a medication event will cover type, name, frequency, dosage, route, etc. It shares the same objective as DescEmb of being agnostic to differing EHR schemas across institutions.

- **Med-BERT** (Rasmy et al., 2021): is a BERT-based foundation model that is pretrained on structured EHR records for disease prediction tasks. It relies solely on structured diagnosis ICD codes as pretraining and evaluation data. The pretraining approach typically follows masked language modeling as proposed in the BERT paper (Devlin et al., 2019).

- **BEHRT** (Li et al., 2020): is another BERT-based foundation model that learns contextual embeddings of the patient's structured records in a similar fashion to Med-BERT via masked language modeling using diagnosis codes. However, it differs from Med-BERT by incorporating patient age as a temporal signal.

- **Hi-BEHRT** (Li et al., 2022): is a hierarchical extension of BEHRT that aims to handle longer, more comprehensive patient records. It captures short- and long-term patient patterns from hierarchically segmented medical sequences and is pretrained with the self-supervised BYOL framework (Grill et al., 2020).

- **CEHR-BERT** (Pang et al., 2021): is another BERT-based model that incorporates temporal and visit information to improve disease prediction. It represents patient visits as sequences of medical codes, augmented with artificial time tokens, age embeddings, and time embeddings.

- **EHRMamba** (Fallahpour et al., 2024): typically follows the embedding structure of CEHR-BERT but uses the state-space architecture MAMBA (Gu & Dao, 2024) instead of a BERT encoder. The use of a state-space model aims to enable efficient processing of relatively long EHR sequences compared to transformer models.

- **REMed** (Kim et al., 2023): is a retrieval-based foundation model that processes textual descriptions in a similar fashion to both DescEmb and GenHPF. It operates by randomly sampling events from patient history and ranking them to retrieve the most relevant ones for different downstream tasks. In contrast to our proposed model, REMed operates at the medical event level, whereas `EHR-RAGp` operates at historical segment level to cover broad history beyond limited set of events, 128 in case of REMed.

# E. Additional Results

### E.1. Ablations

**Effect of Regularization.** Table S6 evaluates the impact of prototype regularization ($\lambda_u$) on both representation behavior and downstream performance. Without regularization, prototype assignments are highly peaked for queries (e.g., $P_{\max} = 0.887$ for Long LOS), indicating collapse to a few dominant prototypes, while history assignments remain sparse and less informative. Enabling regularization significantly increases entropy and reduces $P_{\max}$ for both query and history, leading to more distributed and balanced prototype usage. This results in consistent performance gains across tasks, with Long LOS AUPRC improving from 0.579 to 0.618, and ICU readmission AUPRC from 0.118 to 0.156. Overall, the results show that regularization prevents prototype collapse, promotes better utilization of the prototype space, and improves performance. See Appendix E.4 for further analysis on the usage regularization $\lambda_u$.

**Effect of Chunking Strategies.** Table S5 presents an ablation over chunking strategies. Overall, performance is consistent across all strategies, indicating that `EHR-RAGp` is robust to how patient history is segmented. Event-based chunking achieves the best results for ICU readmission (AUROC 0.747, AUPRC 0.156), while care-stage chunking performs best for in-hospital mortality (AUROC 0.940, AUPRC 0.716). Visit-level chunking yields the strongest performance for long-term

*Table S6.* Ablation of regularization term with different number of prototypes (#p).

| #p | $\lambda_u$ | Query | | History | | Performance | |
|---|---|---|---|---|---|---|---|
| | | P max | Entropy | P Max | Entropy | AUROC | AUPRC |
| | | | | **Long LOS 7d** | | | |
| 64 | × | 0.887 | 0.663 | 0.094 | 4.048 | 0.866 (0.857, 0.876) | 0.579 (0.552, 0.605) |
| | ✓ | 0.084 | 4.096 | 0.027 | 4.156 | 0.884 (0.875, 0.893) | 0.618 (0.590, 0.645) |
| | | | | **ICU-Readmit 30d** | | | |
| 512 | × | 0.716 | 1.578 | 0.030 | 6.099 | 0.730 (0.709, 0.752) | 0.118 (0.098, 0.144) |
| | ✓ | 0.014 | 6.198 | 0.004 | 6.232 | 0.747 (0.724, 0.768) | 0.156 (0.128, 0.189) |

*Table S7.* Ablation of number of prototypes.

| #p | ICU-Readmit 30d | | In-hospital Mortality | |
|---|---|---|---|---|
| | AUROC | AUPRC | AUROC | AUPRC |
| 64 | 0.728 (0.705, 0.751) | 0.132 (0.109, 0.161) | 0.937 (0.931, 0.943) | 0.704 (0.680, 0.726) |
| 128 | 0.736 (0.711, 0.757) | 0.144 (0.121, 0.176) | **0.937 (0.930, 0.943)** | **0.706 (0.683, 0.730)** |
| 256 | 0.739 (0.715, 0.761) | 0.145 (0.121, 0.179) | 0.934 (0.928, 0.94) | 0.695 (0.672, 0.721) |
| 512 | **0.743 (0.720, 0.764)** | **0.150 (0.124, 0.182)** | 0.935 (0.928, 0.941) | 0.697 (0.673, 0.723) |

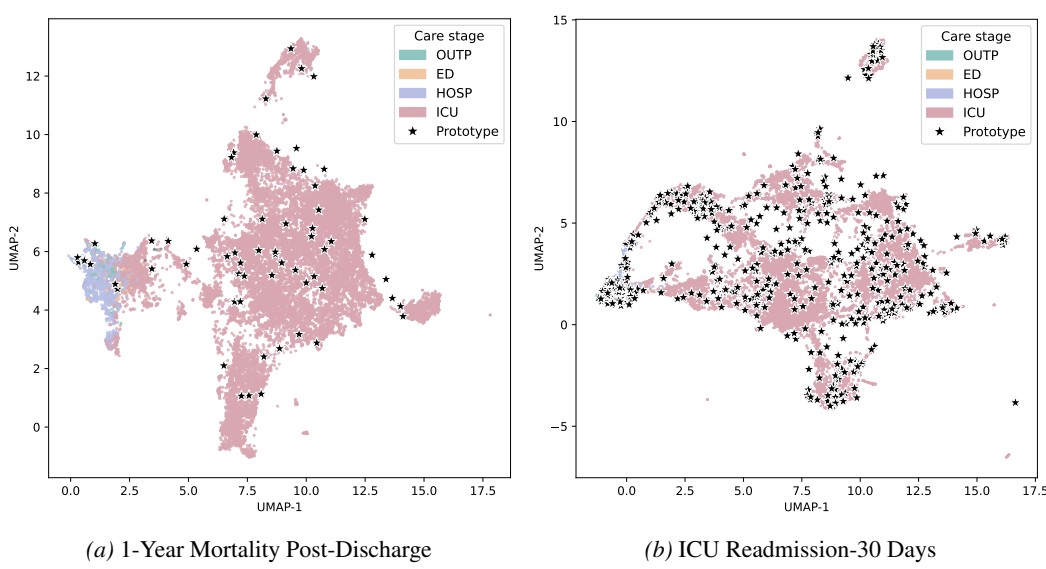

*(a)* 1-Year Mortality Post-Discharge      *(b)* ICU Readmission-30 Days

*Figure S4.* UMAP visualization of history chunk embeddings and prototypes for (a) long length of stay and (b) in-hospital mortality tasks.

outcomes, including long LOS (AUPRC 0.628) and 1-year mortality (AUPRC 0.396). Time-based chunking performs competitively across all tasks but does not outperform other strategies. These results suggest that no single chunking scheme dominates universally, and different chunking methods capture complementary aspects of patient trajectories depending on the prediction task.

**Effect of Prototypes Count.** Table S7 shows the effect of varying the number of prototypes. Increasing the number of prototypes consistently improves performance for ICU readmission, with AUROC rising from 0.728 to 0.743, with 64 and 512 prototypes, and AUPRC from 0.132 to 0.150, indicating that a richer prototype set better captures fine-grained structure in patient trajectories. In contrast, performance on in-hospital mortality remains relatively stable across configurations, with marginal variations around AUROC and AUPRC. Overall, this suggests that the effective number of prototypes is also task dependent.

### E.2. Latent Representation Structure

Figure S4 presents UMAP projections of history chunk embeddings for 1-Year Mortality Post-Discharge and ICU Readmission-30 Days.

### E.3. Qualitative Analysis of Prototype-Guided Retrieval

This section provides a qualitative analysis of the prototype-guided retrieval mechanism in `EHR-RAGp`. To illustrate the behavior of the model, we present two representative case studies from the in-hospital mortality task, corresponding to positive and negative prediction examples. For each case, we analyze prototype usage, query-history alignment, chunk-level relevance weighting, and the contribution of visits and care stages to the final prediction. These examples provide insight into how `EHR-RAGp` organizes longitudinal patient history and selectively integrates clinically relevant context during inference.

#### E.3.1. CASE OF POSITIVE EXAMPLE

Figure S5 presents a representative positive example from the in-hospital mortality task, corresponding to a correctly predicted positive case (`label=1`, `pred=1`) with high confidence (`prob=0.998`) for a query segment originating from Visit V2 in the patient timeline. Panel (a) shows the query prototype distribution across the top activated prototypes, where prototype $P17$ receives the highest assignment probability. Panel (b) illustrates the alignment between retrieved history chunks and the top query prototypes, revealing consistent prototype agreement across several retrieved chunks, particularly among the early ICU-related chunks. Panel (c) shows the final prototype-guided chunk weights, where the model assigns the highest importance to chunks originating from the second visit and predominantly associated with the ICU care stage. Chunks from lower-acuity stages such as ED and outpatient care receive comparatively lower weights. Panels (d) and (e) further aggregate these contributions at the visit and care-stage levels, showing that the prediction is primarily driven by the recent visit trajectory and high-acuity clinical context. Overall, the example illustrates how `EHR-RAGp` selectively emphasizes clinically severe and temporally relevant historical context when forming the final prediction.

#### E.3.2. CASE OF NEGATIVE EXAMPLE

Figure S6 presents a representative negative example from the in-hospital mortality task, corresponding to a correctly predicted negative case (`label=0`, `pred=0`) with low predicted probability (`prob=0.006`) for a query segment originating from Visit V5 in the patient timeline. Panel (a) shows the query prototype distribution across the top activated prototypes, with a more diffuse assignment pattern, indicating less concentration around highly activated latent modes. Panel (b) illustrates the prototype agreement between the query and retrieved history chunks, where alignment is more uniformly distributed across chunks without a dominant subset receiving consistently strong activations. In panel (c), the final prototype-guided chunk weights are relatively balanced across visits and care stages, with no strong emphasis on ICU-related segments. Panels (d) and (e) further confirm this behavior, showing that importance is distributed across multiple visits and care stages rather than concentrated around a single high-acuity trajectory. Overall, this example illustrates how `EHR-RAGp` leverages broader longitudinal history across multiple visits and care stages to support a confident negative prediction, without relying on a single dominant high-acuity trajectory.

### E.4. Analysis of Prototype Regularization

This section analyzes the effect of prototype regularization on the behavior of `EHR-RAGp`. Specifically, we compare the model with and without regularization from two perspectives: (1) global prototype utilization patterns, illustrating how regularization affects the distribution and diversity of prototype assignments, and (2) qualitative case studies, showing how regularization influences prototype-guided retrieval and chunk weighting for individual patient trajectories. Together, these analyses highlight the role of regularization in preventing prototype collapse and encouraging more balanced use of the latent prototype space.

#### E.4.1. GLOBAL PROTOTYPES USAGE

Figure S7 illustrates the effect of prototype regularization on global prototype utilization. Without regularization as in Figure S7a, query assignments exhibit highly imbalanced usage patterns, where a small subset of prototypes receive disproportionately large assignment probabilities while many prototypes remain minimally utilized. A similar, although less severe, imbalance is also observed for history chunk assignments. This behavior indicates partial prototype collapse, where the model concentrates representation learning around a limited number of latent prototypes. In turn, such a situation makes agreement between history and query rather difficult due to the sharper distribution of the query as compared to the history assignments. In contrast, enabling regularization, as in Figure S7b, produces a substantially more uniform assignment distribution across prototypes for both queries and history chunks. The resulting increase in prototype diversity suggests that

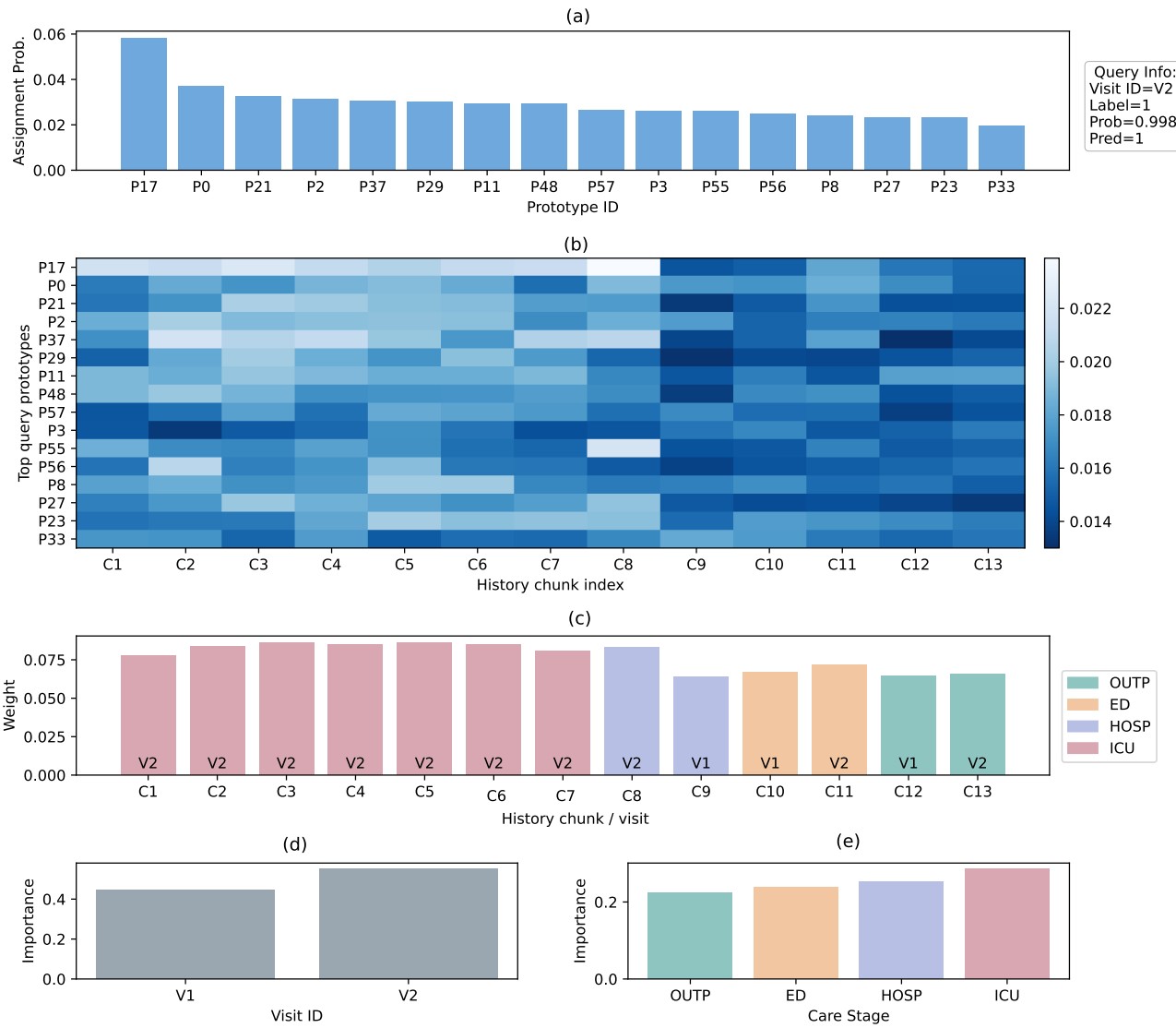

*Figure S5.* Qualitative analysis of prototype-guided retrieval for an in-hospital mortality positive example. (a) Top prototype assignments for the query segment. (b) Alignment between retrieved history chunks and the query prototypes. (c) Prototype-guided relevance weights assigned to retrieved chunks. (d) Aggregated contribution of retrieved visits. (e) Aggregated importance across care stages.

regularization encourages broader utilization of the latent prototype space, leading to more balanced representation learning and improved retrieval behavior.

### E.4.2. QUALITATIVE ANALYSIS OF PROTOTYPES USAGE

To further understand the role of usage regularization $\lambda_u$, we examine the model behavior with regularization, as in in Figure S8, and without regularization, as in Figure S9. In the settings where usage regularization is activated, prototype assignments are distributed across multiple latent prototypes, resulting in relatively balanced alignment scores and chunk weights across retrieved history segments. This behavior indicates that the model integrates information from several relevant historical chunks rather than relying on a single dominant prototype or retrieval path. On the other hand, without usage regularization, the query representation collapses strongly toward a single prototype (P0), which leads to highly concentrated retrieval behavior where few retrieved chunk receives the majority of the final weight while the remaining chunks contribute minimally. With respect to the model performance, it can be observed that despite the collapse the model can still provide correct prediction but less confident. Overall, these examples qualitatively demonstrate how prototype regularization

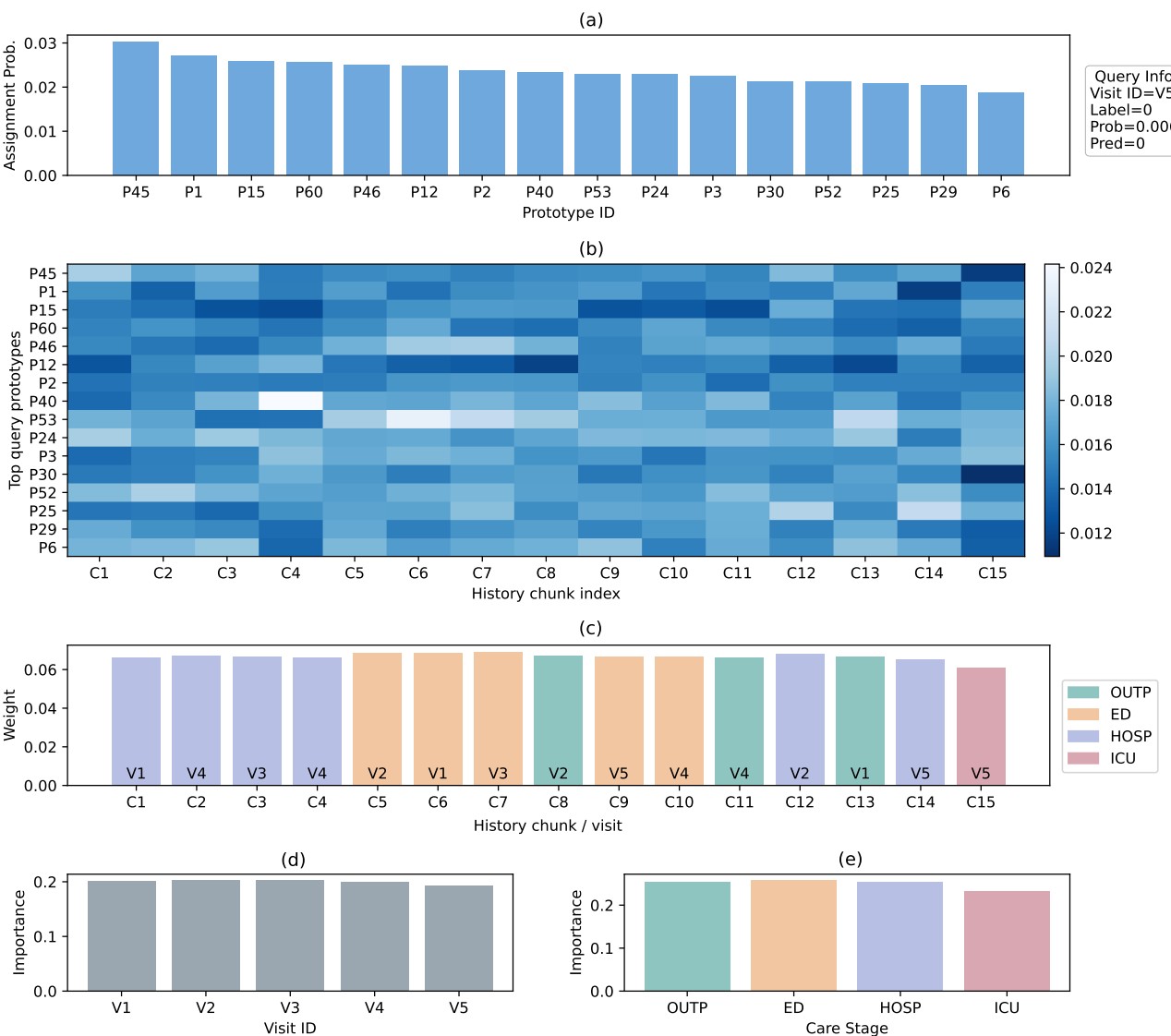

*Figure S6.* Qualitative analysis of prototype-guided retrieval for an in-hospital mortality negative example. (a) Top prototype assignments for the query segment. (b) Alignment between retrieved history chunks and the query prototypes. (c) Prototype-guided relevance weights assigned to retrieved chunks. (d) Aggregated contribution of retrieved visits. (e) Aggregated importance across care stages.

stabilizes retrieval dynamics, encourages broader utilization of patient history, and prevents highly concentrated dependence on a narrow subset of latent prototypes, which in turn deteriorates the retrieval quality.

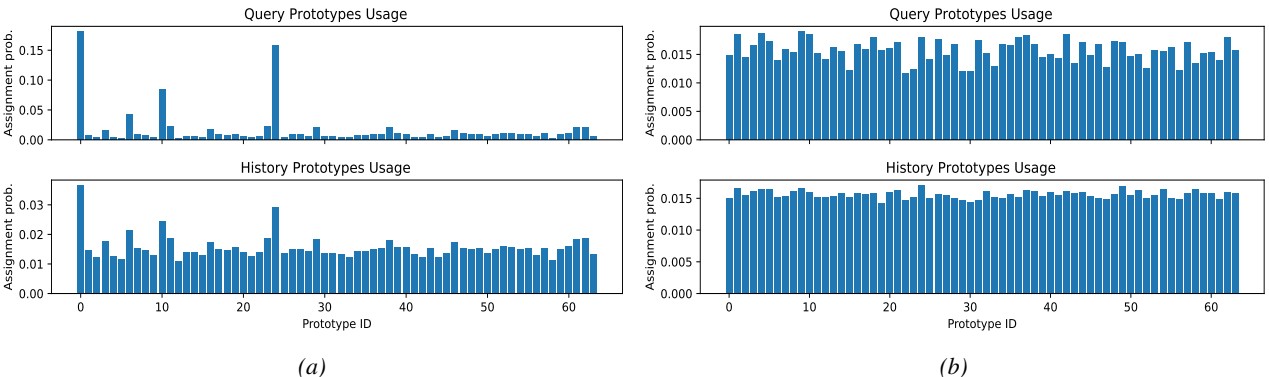

*Figure S7.* Global prototype utilization patterns. (a) Without Regularization. (b) With regularization.

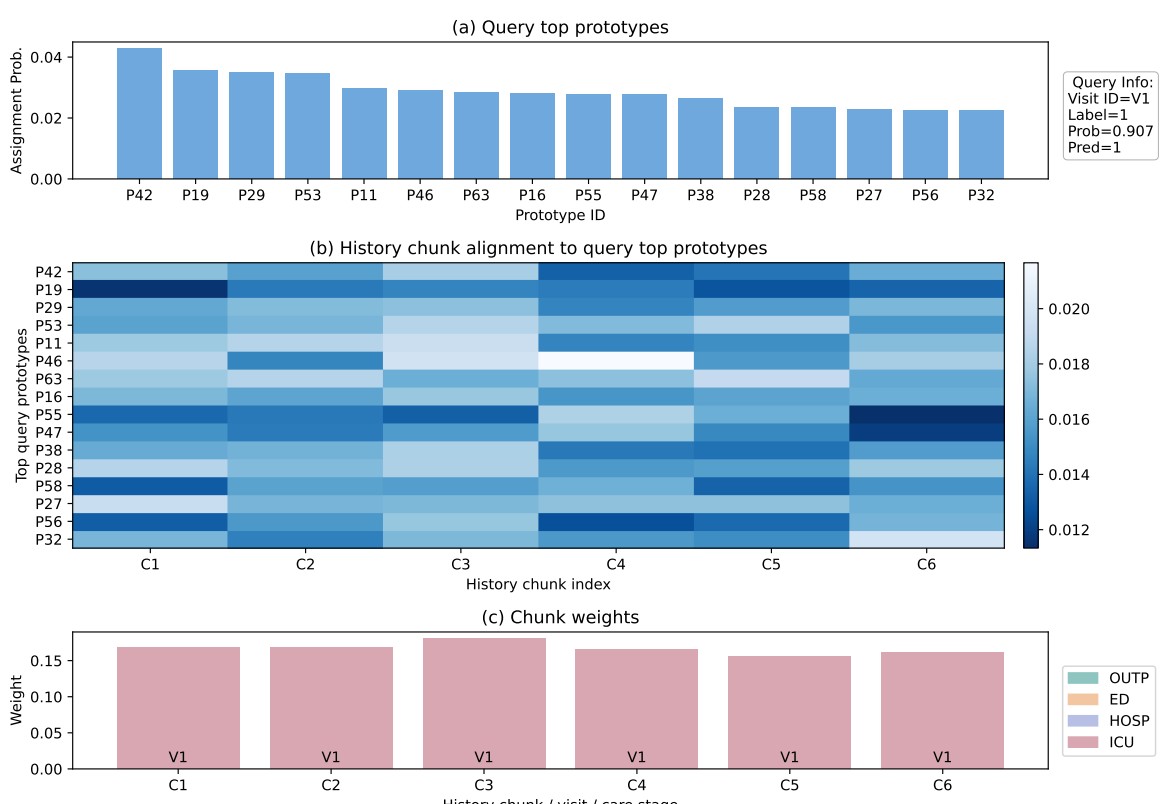

*Figure S8.* Qualitative analysis of sample prediction for long length of stay (7 days) task with regularization (a) Top prototype assignments for the query segment. (b) Alignment between retrieved history chunks and the query prototypes. (c) Prototype-guided relevance weights assigned to retrieved chunks.

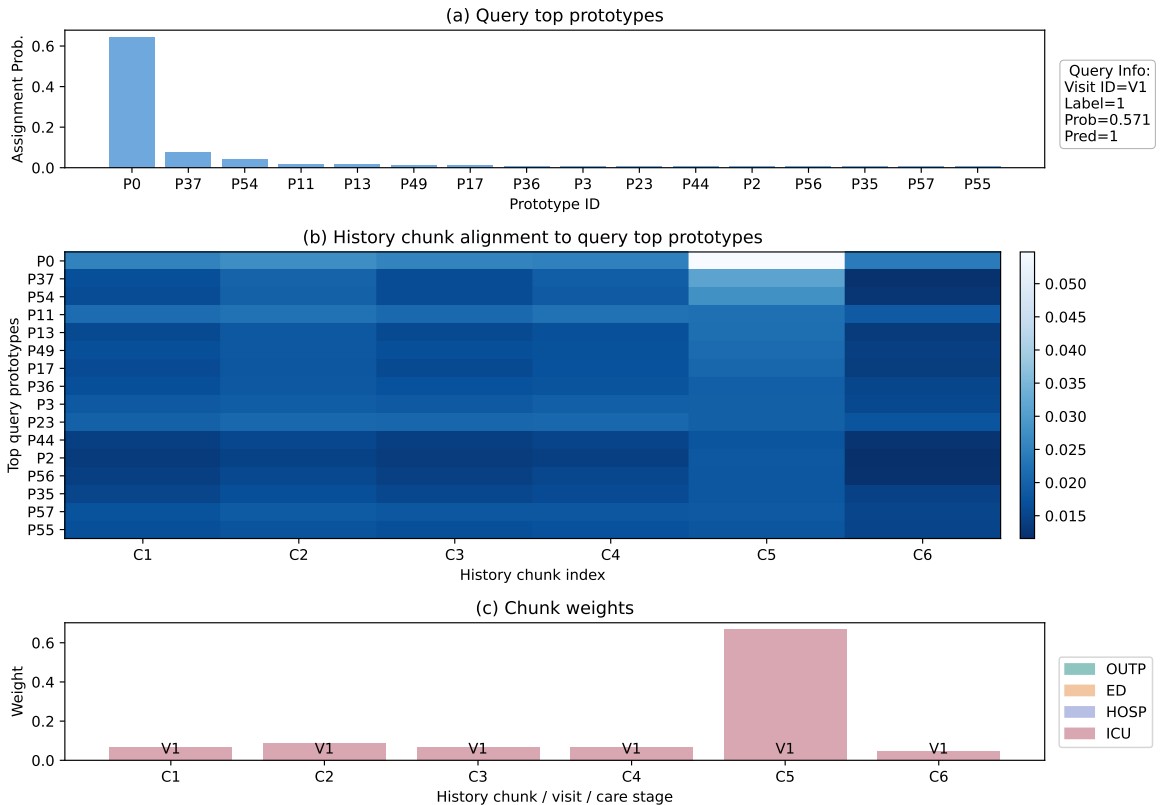

*Figure S9.* Qualitative analysis of sample prediction for long length of stay (7 days) task without regularization (a) Top prototype assignments for the query segment. (b) Alignment between retrieved history chunks and the query prototypes. (c) Prototype-guided relevance weights assigned to retrieved chunks.

