# OpenReview forum: "Retrieval-Augmented Foundation Model Enhances Risk Prediction Using Electronic Health Records"
_ICML.cc/2026/Workshop/FMSD — FMSD @ ICML 2026 Poster_

### Official Review · Reviewer_qa4f · 2026-05-16
**a retrieval-augmented foundation model for electronic health records that addresses the challenge of modeling long patient trajectories**

**Rating:** 5
**Confidence:** 4

**Review:**

pros:

- preprocessing and cohort construction are described in useful detail: the draft specifies MEDS conversion, ICD-9 to ICD-10 mapping, event types, numerical value handling, time-delta encoding, care-stage and visit-order embeddings, patient-level splits, and exclusion of test patients from pretraining. These choices make the experimental setup easier to audit than many EHR modeling papers.

 - main tables cover four clinically common prediction tasks and report both AUROC and AUPRC with bootstrap confidence intervals. The strongest empirical gains are on long length of stay and 1-year mortality, where EHR-RAGp improves over the listed baselines by nontrivial margins in AUPRC.

cons:

 - A central soundness gap is the lack of ablations that isolate retrieval from the backbone, feature representation, and fusion module. The draft needs comparisons against the same RoFormer model with query only, query plus most recent chunks, random chunks, cosine retrieval without prototype weighting, prototype weighting without semantic retrieval, and a matched-token long-context or truncated-history baseline. Without these, the reported gains cannot be attributed specifically to prototype-guided retrieval.

 - claim that the method integrates a patient’s full history is stronger than what is demonstrated. Retrieval always selects a fixed top-M set of 24 chunks, and the paper does not report history length distributions, recall of clinically relevant prior events, performance as a function of available history, or failure cases where important history is not retrieved. The evidence supports selective use of retrieved chunks on MIMIC-IV, not full-history reasoning in a stronger sense.

 - Novelty positioning is incomplete. Related work cites RAM-EHR, EMERGE, KAMELEON, and REMed, but the experiments only include REMed among retrieval-based EHR methods. If the omitted systems are not comparable because they use external knowledge, multimodal inputs, or different task definitions, the paper should state this precisely; otherwise at least one stronger retrieval baseline should be adapted.

 - evaluation is limited to MIMIC-IV adult ICU cohorts. That is a reasonable first setting, but it does not support broad claims about EHR foundation models across healthcare systems, coding practices, or patient populations. External validation, temporal validation, or a clearer scope statement would make the claims better matched to the evidence.

---

### Official Review · Reviewer_4VzT · 2026-05-20
**Solid empirical work but incremental over REMed; ablations don't isolate the prototype contribution**

**Rating:** 6
**Confidence:** 4

**Review:**

## Summary

The authors propose EHR-RAGp, an encoder-based EHR foundation model (RoFormer-base backbone) that augments standard fine-tuning with retrieval over a per-patient vector database of history chunks, plus a learnable set of L prototypes that produces soft relevance weights over the top-M retrieved chunks. The model is pretrained with MLM on MIMIC-IV (199,012 patients) and evaluated on four ICU tasks: in-hospital mortality, 30-day ICU readmission, length of stay (7d), and 1-year post-discharge mortality. Gains are reported against a broad set of baselines (DescEmb variants, GenHPF, Med-BERT, CEHR-BERT, BEHRT, Hi-BEHRT, EHRMamba, REMed). A second result shows that adding EHR-RAGp as a retrieval extension on top of Med-BERT, CEHR-BERT, and BEHRT also improves AUPRC, often by a large margin (e.g., long LOS AUPRC 0.387 to 0.586 for Med-BERT).

## Strengths

- Broad baseline coverage including the closest related work (REMed) and several strong EHR foundation models in a unified vocabulary/embedding setup.
- The plug-in result in Table 2 is the most interesting contribution. The improvements on long LOS and in-hospital mortality AUPRC for Med-BERT and CEHR-BERT are sizeable and suggest the framework has value as a drop-in module, not only as a standalone model.
- Patient-level splits, percentile bootstrap CIs, and Bayesian hyperparameter search are sensible methodological choices.
- The prototype usage regularization (Table S6) is well-motivated and the without/with comparison shows the regularizer doing useful work.

## Areas for improvement

1. **Delta from REMed is under-specified.** The novelty boils down to (a) chunk-level instead of event-level retrieval and (b) prototype-guided relevance weighting. The paper would benefit from a controlled head-to-head where REMed is given the same chunking granularity and prototype budget. As reported, it is not clear whether the gain comes from the prototype module or just from operating at chunk granularity with a stronger pretrained backbone.

2. **Key ablation missing.** I would expect at least three rows: (i) backbone-only fine-tune on the query window, (ii) backbone + top-M retrieval with uniform weights or plain cosine relevance, (iii) full prototype-guided model. Without (ii), the attribution to the prototype module is not earned.

3. **Per-task chunking selection.** Cross-referencing Table 1 against Table S5, the headline "EHR-RAGp (Ours)" row uses Event-Based chunking for ICU-Readmit and Care-Stage chunking for the other three tasks. For Long LOS AUPRC and 1Y mortality AUPRC, Visit-level chunking is actually slightly higher in S5 than what is reported in Table 1. Either way, the protocol for picking the chunking strategy needs to be stated explicitly (val-set selection? fixed per task by clinical rationale?). Right now it reads as per-task tuning that isn't declared.

4. **Single-site evaluation.** Only MIMIC-IV is used, both for pretraining and downstream. The workshop call explicitly highlights generalization, so a transfer experiment to eICU-CRD or a second institution would substantially strengthen the contribution.

5. **CIs overlap with strong baselines on several cells.** For instance ICU-Readmit AUROC 0.747 (0.724, 0.768) vs EHRMamba 0.725 (0.702, 0.749); In-hospital mortality AUROC 0.940 vs CEHR-BERT 0.933. No paired test or DeLong-style comparison is reported, so the "best" claim on AUROC for those tasks is weaker than the prose suggests. AUPRC gains are more convincing.

6. **Closely related RAG-for-EHR work is named but not benchmarked.** RAM-EHR (Xu et al., 2024), EMERGE (Zhu et al., 2024a), REALM (Zhu et al., 2024b), and KAMELEON (Datta et al., 2025) are all in the related work but not in the comparison. RAM-EHR is arguably the closest and should be in the main table or its absence justified.

7. **Computational cost not reported.** Per-patient FAISS indices over 364k patients, with online chunking on retrieval, have non-trivial storage and latency implications. A latency/memory comparison against EHRMamba or Hi-BEHRT (the long-range competitors) would help.

## Detailed comments and questions

- The frozen retriever R_beta is the pretrained backbone before fine-tuning, while f_alpha is fine-tuned. As f_alpha drifts during downstream training, the keys in the index do not. Did the authors try periodic index refresh or contrastive training of R_beta?
- Why is RAG needed at all over simply extending the context window of the same backbone? A baseline of the same RoFormer with longer context (or sliding-window long-context fine-tuning) would clarify whether the gain comes from retrieval specifically or just from access to more history.
- The 1Y mortality label assignment described in Appendix C.2 (positive label only to stays within one year of the mortality date) risks introducing label leakage across multiple stays of the same patient. Some discussion would help.
- Minor EHRSHOT/Wornow et al. is in citations but not used.

## Justification of score

The empirical study is broad and the Table 2 plug-in result is a real and useful finding. However, the contribution over REMed/RAM-EHR is incremental in framing, the ablations as written do not isolate the effect of the prototype module from the effect of retrieval, the headline numbers in Table 1 appear to use per-task chunking choices without a stated selection protocol, and the evaluation is limited to one dataset. These add up to a paper that I find marginally above the bar in its current form. Adding the missing ablation (retrieval without prototypes), making the chunking-selection protocol explicit, and either a second-site transfer or RAM-EHR/EMERGE in the comparison would move me to a clear accept.